# The impact of the COVID-19 lockdown on birthweight among singleton term birth in Denmark

Victoria E. de Knegt[1,2,3], Paula L. Hedley[1,4], Gitte Hedermann[1,5], Casper Wilstrup[1,6], Marie Bækvad-Hansen[1,7], Ida N. Thagaard[8], Henrik Hjalgrim[9,10,11], Jørgen Kanters[12], Mads Melbye[3,11,13], David M. Hougaard[1,7], Anders Hviid[9,14], Lone Krebs[3,15], Morten Breindahl[16], Ulrik Lausten-Thomsen[16☯*], Michael Christiansen[1,12☯*]

1 Department for Congenital Disorders, Statens Serum Institut, Copenhagen, Denmark, 2 Department of Paediatrics, Copenhagen University Hospital Slagelse, Slagelse, Denmark, 3 Department of Clinical Medicine, University of Copenhagen, Copenhagen, Denmark, 4 Brazen Bio, Los Angeles, California, United States of America, 5 Department of Obstetrics and Gynaecology, Copenhagen University Hospital, Rigshospitalet, Copenhagen, Denmark, 6 Abzu, Copenhagen, Denmark, 7 Danish Center for Neonatal Screening, Statens Serum Institut, Copenhagen, Denmark, 8 Department of Obstetrics and Gynaecology, Copenhagen University Hospital Slagelse, Slagelse, Denmark, 9 Department of Epidemiology Research, Statens Serum Institut, Copenhagen, Denmark, 10 Department of Haematology, Copenhagen University Hospital Rigshospitalet, Copenhagen, Denmark, 11 Danish Cancer Society Research Center, Copenhagen, Denmark, 12 Department of Biomedical Sciences, University of Copenhagen, Copenhagen, Denmark, 13 Department of Medicine, Stanford University School of Medicine, Stanford, CA, United States of America, 14 Pharmacovigilance Research Center, Department of Drug Development and Clinical Pharmacology, Faculty of Health and Medical Sciences, University of Copenhagen, Copenhagen, Denmark, 15 Department of Obstetrics and Gynaecology, Copenhagen University Hospital, Amager and Hvidovre Hospital, Copenhagen, Denmark, 16 Department of Neonatology, Copenhagen University Hospital Rigshospitalet, Copenhagen, Denmark

☯ These authors contributed equally to this work.
* mic@ssi.dk (MC); ulrik.lausten-thomsen@regionh.dk (ULT)

## Abstract

In Denmark, a nationwide COVID-19 lockdown was implemented on March 12, 2020 and eased on April 14, 2020. The COVID-19 lockdown featured reduced prevalence of extremely preterm or extremely low birthweight births. This study aims to explore the impact of this COVID-19 lockdown on term birthweights in Denmark. We conducted a nationwide register-based cohort study on 27,870 live singleton infants, born at term (weeks 37–41), between March 12 and April 14, 2015–2020, using data from the Danish Neonatal Screening Biobank. Primary outcomes, corrected for confounders, were birthweight, small-for-gestational-age (SGA), and large-for-gestational-age (LGA), comparing the COVID-19 lockdown to the previous five years. Data were analysed using linear regression to assess associations with birthweight. Multinomial logistic regression was used to assess associations with relative-size-for-gestational-age (xGA) categories. Adjusted mean birthweight was significantly increased by 16.9 g (95% CI = 4.1–31.3) during the lockdown period. A dip in mean birthweight was found in gestational weeks 37 and 38 balanced by an increase in weeks 40 and 41. The 2020 lockdown period was associated with an increased LGA prevalence (aOR 1.13, 95% CI = 1.05–1.21). No significant changes in proportions of xGA groups were found between 2015 and 2019. The nationwide COVID-19 lockdown resulted in a

**Data Availability Statement:** Relevant data are within the paper. Full data cannot be shared publicly due to data protection regulations. Data are available from Statens Serum Institut (contact

either via corresponding author MC, mic@ssi.dk, or directly via the institution, serum@ssi.dk) for researchers who meet the criteria for access to confidential data.

**Funding:** The author(s) received no specific funding for this work.

**Competing interests:** Dr Breindahl has a patent (NeoHelp) with royalties paid. Dr Breindahl has nothing else to disclose. All other authors have nothing else to disclose. This does not alter our adherence to PLOS ONE policies on sharing data and materials.

small but significant increase in birthweight and proportion of LGA infants, driven by an increase in birthweight in gestational weeks 40 and 41.

## Introduction

SARS-CoV-2 infections in pregnancy are associated with adverse pregnancy outcomes [1]. However, SARS-CoV-2 mitigation efforts in some high-income countries showed decreased preterm and extreme preterm birth rates not caused by increased stillbirth and perinatal death rates [2–7]. The COVID-19 pandemic has thus both direct–related to SARS-CoV-2 infection —and indirect effects on maternal and foetal health. The indirect consequences of the COVID-19 pandemic may be substantial [2, 7], as all pregnant women, infected and non-infected alike, have experienced the implementation of strict COVID-19 mitigation strategies.

Both increased and decreased birth weight (BW) have been linked to a range of adverse outcomes during birth, in childhood and in later life [8], including hypertension and other chronic diseases [9, 10]. BW and the relative-size-for-gestational-age-BW (xGA) (e.g. large-for-gestational-age (LGA) or small-for-gestational-age (SGA)) are frequently used as surrogate markers of intrauterine conditions [11]. Any detectable changes in term BW would indicate that COVID-19 lockdown and other indirect consequences of the COVID-19 pandemic have affected prenatal growth, which may affect future health.

Globally, the COVID-19 pandemic was mitigated through restrictions in social interaction, workplace attendance, school attendance, and travel, as well as containment policies for SARS-CoV-2 exposed individuals [2, 12]. The COVID-19 mitigation in Denmark was at its strictest in the period from March 12 to April 14, 2020.

It is important to assess the consequences of pandemic mitigation policies as they may profoundly influence care-seeking behaviour and access to healthcare [13, 14]. The aim of this study was to investigate the indirect effects of the COVID-19 pandemic on maternal and infant health by examining BW and xGA distribution during the initial lockdown implemented in Denmark compared to the previous five years.

## Methods

### Study design and participants

We performed a nationwide cohort study, including 27,870 singleton neonates for whom data was available from the Danish Neonatal Screening Biobank (DNSB). In Denmark it is mandatory to offer neonatal screening within three days to all newborns, and the uptake is nearly 100% [3]. All included infants were born at term (gestational age (GA) $37^{+0}$–$41^{+6}$) from March 12 to April 14, 2015–2020. We excluded births in which BW or gestational age data were missing or believed to be likely due to registration errors (above or below the 5*SD of the mean for each gestation week for each sex). When the number of term pregnancies obtained from the DNSB in this study was compared to the total number of term pregnancies registered in the Medical Birth Registry and The Danish National Patient Registry [2], the proportion excluded was 3.6%. BW was recorded in grams (g). Length of gestation, calculated from an estimated due date determined by ultrasound in first trimester, was obtained from the DNSB and reported in number of completed weeks. We defined SGA as BW less than 10th percentile and LGA greater than the 90th percentile at each completed week, for each sex, according to international standards of newborn size [15]. As the proportion of Danish babies with LGA is > 25% using the international standard, we also defined SGA and LGA as births with BW,

corrected for sex and gestational week, below the 5th percentile or above the 95th percentile of the study population from 2015–2019, respectively.

## Statistical analyses

The relationship between BW and period of birth (2015–2019 vs 2020), length of gestation, and sex was assessed using multivariable linear regression. The relationships between LGA or SGA and period of birth, length of gestation, and sex was assessed using multivariable logistic regression. Odds ratios (OR) were adjusted for sex and week of gestation at birth. The R libraries stats and nnet and R vers 4.2.0 were used.

## Ethics approval

The study was conducted according to Danish legislation and guidelines for register research and was approved by the Data Protection Agency officer at Statens Serum Institut (No: 20/04753).

## Results

The distribution of singleton births in each gestation week and xGA category for each year, as well as the distribution of mean BW for these categories for the period of March 12 to April 14, 2015–2019 and 2020 are shown in Table 1. No significant temporal trends in birth rate or sex ratio were noted during the period 2015–2019; the highest ratio of female births occurred in 2018 (49.6% females), the lowest ratio of female births occurred in 2016 (48.3% females).

The proportion of births in weeks 40 and 41 was 59.7% and significantly higher in the lockdown period compared to 57.7% (95% CI 57.5, 57.9) in the previous five years (Table 1). This was not the result of a trend through previous years as the gestational week distribution was not significantly associated with year of birth for births between 2015 and 2019 ($\chi^2$ test of equal proportions, p = 0.71).

### Birthweight

The crude mean BW among all term neonates increased by 26.4 g (95% CI 11.1, 41.7) in the lockdown period (March 12 to April 14, 2020) compared to the previous five years (Table 2). As expected, mean BW was significantly associated with sex and length of gestation. After adjustment for these confounders, the increase in adjusted BW during the 2020 lockdown period was 16.9 g (95% CI 2.9, 30.7) (Table 2).

As the length of gestation was clearly associated with BW, we assessed whether this association differed between early term births (week 37–38) and late term births (week 39–41). The results of two multivariable linear regressions–one for each period–(Table 2) showed that the increase in BW associated with birth during the lockdown period was driven by a positive association in week 39–41.

### Size-for-gestational-age (xGA)

Using the INTERGROWTH-21 definition of LGA and SGA, the LGA proportion increased for both sexes during the 2020 lockdown period compared to the calendar matched period for the previous five years (girls: 28.9% vs 26.7% (95% CI 25.1, 28.3), and boys: 29.8% vs 26.8% (95% CI 25.7, 28.0)) (Table 1). The sex and gestational week adjusted OR for LGA was 1.13 (95% CI 1.05, 1.21) in the lockdown period (Table 3). Neither LGA (β = 0.009 95% CI -0.002, 0.020) nor SGA (β = -0.001 95% CI -0.005, 0.004) ORs were significantly increased with year during the period 2015–2019. Using the population-based reference with

**Table 1. Distribution of births per gestation week and xGA category for each year.** Comparison of mean birth weights among the 2015–2019 aggregate data and the 2020 lockdown period, March 12 to April 14.

| girls | 2015 | | 2016 | | 2017 | | 2018 | | 2019 | | 2020 | | 2015–2019 | | 2020 | |
|---|---|---|---|---|---|---|---|---|---|---|---|---|---|---|---|---|
| | N | | N | | N | | N | | N | | N | | mean birth weight, g (SD) | | | |
| SGA | 68 | 3.2% | 58 | 2.6% | 52 | 2.2% | 72 | 3.0% | 58 | 2.6% | 58 | 2.6% | 2506.1 | (254.88) | 2430.1 | (222.66) |
| AGA | 1532 | 72.3% | 1591 | 70.6% | 1652 | 70.9% | 1672 | 69.6% | 1561 | 69.6% | 1550 | 68.5% | 3323.8 | (306.42) | 3331.8 | (299.52) |
| LGA | 520 | 24.5% | 604 | 26.8% | 626 | 26.9% | 659 | 27.4% | 623 | 27.8% | 654 | 28.9% | 4040.3 | (301.95) | 4059.0 | (300.63) |
| | N | | N | | N | | N | | N | | N | | mean birth weight, g (SD) | | | |
| week 37 | 86 | 4.1% | 107 | 4.7% | 113 | 4.8% | 120 | 5.0% | 103 | 4.6% | 95 | 4.2% | 3029.2 | (464.35) | 3040.0 | (510.73) |
| week 38 | 264 | 12.5% | 316 | 14.0% | 280 | 12.0% | 323 | 13.4% | 295 | 13.2% | 274 | 12.1% | 3247.6 | (451.11) | 3178.0 | (419.09) |
| week 39 | 513 | 24.2% | 531 | 23.6% | 563 | 24.2% | 562 | 23.4% | 513 | 22.9% | 507 | 22.4% | 3396.5 | (414.14) | 3421.3 | (426.55) |
| week 40 | 703 | 33.2% | 694 | 30.8% | 758 | 32.5% | 815 | 33.9% | 731 | 32.6% | 754 | 33.3% | 3548.3 | (423.91) | 3583.7 | (422.93) |
| week 41 | 554 | 26.1% | 605 | 26.9% | 616 | 26.4% | 583 | 24.3% | 600 | 26.8% | 632 | 27.9% | 3717.0 | (434.82) | 3739.8 | (441.31) |
| TOTAL | 2120 | 100.0% | 2253 | 100.0% | 2330 | 100.0% | 2403 | 100.0% | 2242 | 100.0% | 2262 | 100.0% | 3493.0 | (468.04) | 3519.0 | (477.07) |
| boys | 2015 | | 2016 | | 2017 | | 2018 | | 2019 | | 2020 | | 2015–2019 | | 2020 | |
| | N | | N | | N | | N | | N | | N | | mean birth weight, g (SD) | | | |
| SGA | 59 | 2.7% | 69 | 2.9% | 52 | 2.1% | 59 | 2.4% | 67 | 2.8% | 60 | 2.5% | 2614.5 | (256.79) | 2600.1 | (263.77) |
| AGA | 1545 | 71.2% | 1713 | 71.0% | 1701 | 69.6% | 1724 | 70.5% | 1681 | 70.5% | 1627 | 67.7% | 3444.9 | (319.64) | 3438.0 | (318.25) |
| LGA | 565 | 26.0% | 630 | 26.1% | 692 | 28.3% | 662 | 27.1% | 638 | 26.7% | 716 | 29.8% | 4183.2 | (317.26) | 4205.0 | (310.07) |
| | N | | N | | N | | N | | N | | N | | mean birth weight, g (SD) | | | |
| week 37 | 115 | 5.3% | 129 | 5.3% | 135 | 5.5% | 128 | 5.2% | 126 | 5.3% | 105 | 4.4% | 3126.6 | (473.56) | 3122.2 | (519.23) |
| week 38 | 305 | 14.1% | 344 | 14.3% | 353 | 14.4% | 345 | 14.1% | 331 | 13.9% | 340 | 14.1% | 3371.0 | (463.57) | 3343.3 | (475.20) |
| week 39 | 521 | 24.0% | 565 | 23.4% | 569 | 23.3% | 585 | 23.9% | 582 | 24.4% | 560 | 23.3% | 3550.5 | (431.86) | 3577.3 | (442.86) |
| week 40 | 695 | 32.0% | 715 | 29.6% | 749 | 30.6% | 744 | 30.4% | 747 | 31.3% | 746 | 31.0% | 3688.9 | (440.69) | 3722.4 | (429.39) |
| week 41 | 533 | 24.6% | 659 | 27.3% | 639 | 26.1% | 643 | 26.3% | 600 | 25.1% | 652 | 27.1% | 3847.0 | (437.21) | 3858.4 | (478.97) |
| TOTAL | 2169 | 100.0% | 2412 | 100.0% | 2445 | 100.0% | 2445 | 100.0% | 2386 | 100.0% | 2403 | 100.0% | 3622.0 | (483.52) | 3645.6 | (498.66) |
| ALL (% girls) | 4289 | (49.4%) | 4665 | (48.3%) | 4775 | (48.8%) | 4848 | (49.6%) | 4628 | (48.4%) | 4665 | (48.5%) | 3558.9 | (480.35) | 3584.2 | (492.35) |

AGA: appropriate-for-gestational-age; LGA: large-for-gestational-age; SD: standard deviation; SGA: small-for-gestational-age

LGA $\geq 95^{th}$ percentile BW and SGA $\leq 5^{th}$ percentile BW, the LGA proportion in the lockdown period was also found elevated, OR: 1.16 (95% CI 1.01, 1.33), whereas the SGA proportion was not (Table 3). The LGA OR increased slightly, β = 0.003 (95% CI 0.000, 0.006), per year in the period 2015–2019, but this could not explain the increase seen during the lockdown. The OR for SGA did not change with year during 2015–2019, β = 0.000 (95% CI -0.006, 0.006).

**Table 2. Change in infant birth weight in relation to lockdown, sex, and length of gestation.**

| | Birth weight (grams) increase (95% CI) | | |
|---|---|---|---|
| | Crude[a] | | Adjusted[b] |
| Birth in lockdown period | 26.4 (11.15–41.05) | | 16.9 (2.80–30.75) |
| | Adjusted# BW (grams) increase (95% CI) | | |
| | Gestational week 37–38 | | Gestational week 39–41 |
| Birth in lockdown period | - 33.9 (-68.53–0.69) | | 26.3 (11.33–41.31) |

[a]Linear regression.
[b]Linear regression with adjustment for infant sex and length of gestation.
CI: confidence interval

**Table 3. Sex and gestational age adjusted odds-ratios for LGA and SGA using INTERGROWTH-21 and population-based birth weight references.**

| | INTERGROWTH-21 | | | |
|---|---|---|---|---|
| | LGA | | SGA | |
| **Variable** | **aOR** | **95% CI** | **aOR** | **95% CI** |
| Birth in lockdown period | 1.13 | (1.05–1.21) | 1.00 | (0.82–1.22) |
| | Population-based | | | |
| | LGA ($\geq$ 95<sup>th</sup> percentile BW) | | SGA ($\leq$ 5<sup>th</sup> percentile BW) | |
| Birth in lockdown period | 1.16 | (1.01–1.33) | 0.99 | (0.85–1.15) |

aOR: adjusted odds ratio; BW: birth weight, CI: confidence interval; LGA: large-for-gestational-age; SGA: small-for gestational-age

## Discussion

We found that giving birth at term, during the Danish national lockdown period (March 12 to April 14, 2020), was significantly associated with increased BW compared to the five previous years. This result is compatible with two previous local studies; an Austrian study examining a variety of adverse pregnancy outcomes observed a significantly higher mean BW ($\approx$35g) of all infants (i.e. preterm and term) born during the lockdown phase compared to the previous 15 years [16]. They reported that the primary driver of the higher mean BW was an increase in maternal gestational weight gain during the mitigation period. A second study carried out in Wuhan, China found that term infants born during the pandemic were significantly heavier ($\approx$35g) compared to pre-pandemic infants [17]. Maternal body mass index (BMI), caesarean delivery numbers, and proportions of preterm and term neonates were not significantly different between the two groups. The authors suggested that nutritional changes and lack of exercise may explain the increased BW. Both studies reported crude mean BW increases, best compared to the observed unadjusted 26.4 g found in this study. Thus, three studies, two European and one Asian, report term BW increases of a similar magnitude during the early part of the COVID-19 pandemic.

The observed 26.4 g increase in mean BW during lockdown is equivalent to less than a day of intrauterine growth in term infants. On the one hand, this small increase may not seem to be large enough to have any important consequences on infant metabolic health but, on the other hand, the 26.4 g increase is roughly the equivalent of $\approx$10% of the effect of not regularly smoking during pregnancy compared to smoking 6–10 cigarettes per day [18]. Ultimately, the consequences of the observed increase in mean BW depends on whether the increase is the result of pathologic mechanisms, such as increased gestational weight gain during pregnancy, increased incidence of dysregulated gestational diabetes mellitus (GDM), and unbeneficial changes in induction of labour and delivery mode practices, or whether the increase is the result of advantageous factors such as improved maternal mental and physical health, better nutrition, and mothers naturally carrying longer to term.

A significant increase in the proportion of LGA infants born during the lockdown was found compared to the previous five years. Irrespective of whether the INTERGROWTH-21 reference, that is international and results in a LGA proportion of > 25% among Danish children, or the 95<sup>th</sup> percentile population reference was used. This, to our knowledge, is the first observation of increased proportion of LGA infants during COVID-19 mitigation. This increase is worrying as LGA predisposes to delivery complications [19, 20], neonatal hypoglycaemia [21], and obesity and metabolic syndrome later in life [9]. The proportion of LGA infants should be monitored carefully during future pandemic mitigations.

Factors that could result in increased number of LGA infants are increased maternal BMI and gestational weight gain, which are known to affect neonatal growth, particularly in second- and third-trimester [22]. Increased maternal BMI does not, however, explain the (insignificant) dip in BW seen in gestational week 37 and 38 (Table 2). Another theory could be the reduced stress level when staying home causing an increase in BW. A recent randomized trial reported that mindfulness therapy in pregnancy reduced corticosteroid and stress levels in pregnant women causing an increase in BW [23].

We found a modest increase in the proportion of births occurring in weeks 40 and 41 in the lockdown period (Table 1). This raises the question of whether acute changes in delivery practices, with postponement or suspension of labour induction and/or changes in mode of delivery at hospitals led to fewer iatrogenic births and, thus, later delivery. Since BW depends largely on the length of gestation, both scenarios would result in higher BW.

As Denmark provides universal health care including extensive prenatal, childbirth and postnatal services, it is unlikely that changes in access to medical care was a major contributing factor in our setting. However, changes in the allocation of resources within the healthcare sector and disruptions and/or interruptions in antenatal care, with the acute shift from in-house check-ups to virtual controls, may have harmed prophylactic measures and treatment of pregnancy related diseases, such as GDM [23]. Internationally, this effect was evident in the increased number of stillbirths noted in a London hospital during the COVID-19 pandemic, where none of the pregnant women were infected with SARS-CoV-2 [24]. However, a Danish national study found a significant reduction in stillbirth rate during prolonged COVID-19 mitigation following the strict lockdown [2].

Care-seeking behaviour, both decreased [13] and increased [14], may also have affected pregnancy outcomes. Reduced social interactions, physical distancing, better hygiene, and the dramatic fall in overall infection rates of non-COVID-19 infectious diseases [25–27], albeit not chlamydia [29], could all have influenced the overall inflammatory state of pregnant women, and subsequently influenced term BW through yet to be established pathways. Furthermore, factors relating to maternal stress may also have contributed, as stress is a known negative influencer of intrauterine growth [28]. A pronounced increase in depression and anxiety were noted in pregnant women during the lockdown period [14, 29]. Conversely, a reduction in work-related stress, physical work strain, and harmful work- and environmental exposures [30], as well as the ability to rest more while in lockdown, could have had an opposite and positive effect on intrauterine growth and/or prolong gestation. Finally, lifestyle changes, including changes in dietary and exercise habits, resulting in greater gestational weight gain, could have increased term BW.

Our study has several strengths. We have examined a national cohort of 27,870 infants, where data is available from the national neonatal screening program, which has a nearly 100% uptake rate. Neonatal screening is performed within three days of birth at the Danish National Screening Biobank, and relevant birth-related data is registered, based on reliable, real-time, mandatory reporting. Because exposure (the lockdown) is independent of the recorded outcome, differential misclassification is not expected to be a problem. Furthermore, our approach of validating our findings using the international reference from the INTER-GROWTH-21[st] Project by also referring to 95-percentile values for SGA and LGA in our own population addresses any issues concerning the lack of use of more local/Danish growth curves.

However, the study does have some limitations. The use of data from neonatal screening databases is slightly biased towards healthy, uncomplicated births, as stillbirths and very early postnatal deaths (before day three) will not be reflected in the database. Yet, as most early neonatal demise occurs among very preterm babies, it is unlikely to influence the findings of this

study substantially. Furthermore, due to a lack of access to more precise data at the time of analysis, length of gestation was reported in complete weeks rather than days. Thus, we cannot confidently conclude whether length of gestation changed within a period of less than seven days. Finally, we did not have any data on maternal BMI, gestational weight gain, pregnancy complications, labour induction, or mode of delivery, all of which would have aided in identifying a causal link between the observed increased mean BW and the nationwide lockdown. Such data will only become available with time.

## Conclusion

We found that term birth during the nationwide lockdown resulted in increased mean BW and a significant increase in proportion of LGA infants, driven largely by an increase in BW in gestational week 40 and 41. Pandemic mitigation measures should be carefully monitored with respect to effects on perinatal health.

## Acknowledgments

This research was conducted using the Danish Neonatal Screening Biobank and the Danish National Biobank resources.

## Author Contributions

**Conceptualization:** Paula L. Hedley, Gitte Hedermann, Ulrik Lausten-Thomsen, Michael Christiansen.

**Data curation:** Marie Bækvad-Hansen.

**Formal analysis:** Victoria E. de Knegt, Paula L. Hedley, Gitte Hedermann, Marie Bækvad-Hansen, Ida N. Thagaard, Henrik Hjalgrim, Jørgen Kanters, Mads Melbye, David M. Hougaard, Anders Hviid, Lone Krebs, Morten Breindahl, Ulrik Lausten-Thomsen, Michael Christiansen.

**Investigation:** Casper Wilstrup.

**Methodology:** Henrik Hjalgrim.

**Supervision:** Michael Christiansen.

**Writing – original draft:** Victoria E. de Knegt, Paula L. Hedley, Ulrik Lausten-Thomsen, Michael Christiansen.

**Writing – review & editing:** Victoria E. de Knegt, Paula L. Hedley, Casper Wilstrup, Marie Bækvad-Hansen, Henrik Hjalgrim, Anders Hviid, Ulrik Lausten-Thomsen, Michael Christiansen.

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
