## [Decision Letter · Decision Letter 0]

3 Feb 2023

PONE-D-22-31109

The impact of the COVID-19 lockdown on birthweight among singleton term birth in DenmarkThe impact of the COVID-19 lockdown on birthweight among singleton term birth in Denmark

PLOS ONE

Dear Dr. Lausten-Thomsen,

Thank you for submitting your manuscript to PLOS ONE. After careful consideration, we feel that it has merit but does not fully meet PLOS ONE’s publication criteria as it currently stands. Therefore, we invite you to submit a revised version of the manuscript that addresses the points raised during the review process.

We look forward to receiving your revised manuscript.

Kind regards,

Francesca Crovetto

Academic Editor

PLOS ONE

Journal Requirements:

2. For studies reporting research involving human participants, PLOS ONE requires authors to confirm that this specific study was reviewed and approved by an institutional review board (ethics committee) before the study began. Please provide the specific name of the ethics committee/IRB that approved your study, or explain why you did not seek approval in this case.

“I have read the journal's policy and the

authors of this manuscript have the following competing interest: Dr Breindahl has a patent (NeoHelp) with royalties paid. Dr Breindahl has nothing else to disclose.

All other authors reported to have nothing to disclose.”

“This research was conducted using the Danish Neonatal Screening Biobank and the Danish National Biobank resource, funded by the Novo Nordisk Foundation.”

7. Your ethics statement should only appear in the Methods section of your manuscript. If your ethics statement is written in any section besides the Methods, please delete it from any other section.

Reviewers' comments:

Reviewer's Responses to Questions

**Comments to the Author**

1. Is the manuscript technically sound, and do the data support the conclusions?

Reviewer #1: Yes

2. Has the statistical analysis been performed appropriately and rigorously? 

Reviewer #1: Yes

3. Have the authors made all data underlying the findings in their manuscript fully available?

Reviewer #1: Yes

4. Is the manuscript presented in an intelligible fashion and written in standard English?

Reviewer #1: Yes

5. Review Comments to the Author

Reviewer #1: I've read with interest the paper by De Knegt et al. about the impact of the COVID-19 lockdown on birthweight among singleton term birth in Denmark. The study follows the scientific rigor and the manuscript is well written.

I have only one comment, Would the results be the same if the authors use local birthweight charts?

6. PLOS authors have the option to publish the peer review history of their article (what does this mean?). If published, this will include your full peer review and any attached files.

Reviewer #1: No

---

## [Author Response · Author response to Decision Letter 0]

13 Feb 2023

Title: The impact of the COVID-19 lockdown on birthweight among singleton term birth in Denmark 

Manuscript ID: PONE-D-22-31109 

Dear PLOS ONE, Reviewers, and Editors

Thank you for considering our manuscript for publication and giving us an opportunity to submit a revised version of out manuscript. You will find our response to journal requirements (point 1-8) and to reviewer #1 (point 1-6) below. 

Response to Journal requirements (point 1-8): 

-> Response: The manuscript now meets PLOS ONES’s style requirements. The manuscript fulfills the advised templates. 

2. For studies reporting research involving human participants, PLOS ONE requires authors to confirm that this specific study was reviewed and approved by an institutional review board (ethics committee) before the study began. Please provide the specific name of the ethics committee/IRB that approved your study, or explain why you did not seek approval in this case.

-> Response: The study was conducted according to Danish legislation and guidelines for register research stating that anonymized register-based research does not require individual patient consent or ethical approval. This study was accordingly approved be the Data Protection Agency Officer at Statens Serum Institut (No: 20/04753).

-> Response: The study was conducted according to Danish legislation and guidelines for register research stating that anonymized register-based research does not require individual patient consent or ethical approval. This study was accordingly approved be the Data Protection Agency Officer at Statens Serum Institut (No: 20/04753). 

“I have read the journal's policy and the authors of this manuscript have the following competing interest: Dr Breindahl has a patent (NeoHelp) with royalties paid. Dr Breindahl has nothing else to disclose.

All other authors reported to have nothing to disclose.”

-> Response: The statement ‘This does not alter our adherence to PLOS ONE policies on sharing data and materials’ has been added to both the cover letter and the competing interest statement. 

“This research was conducted using the Danish Neonatal Screening Biobank and the Danish National Biobank resource, funded by the Novo Nordisk Foundation.”

-> Response: The funding information has been removed from the acknowledgements. We do not wish to add anything to our funding statement. 

-> Response: Relevant data are within the manuscript. Full data cannot be shared publicly due to data protection regulations. Data are available from Statens Serum Institut (contact via corresponding author MC, mic@ssi.dk) for researchers who meet the criteria for access to confidential data.

7. Your ethics statement should only appear in the Methods section of your manuscript. If your ethics statement is written in any section besides the Methods, please delete it from any other section.

-> Response: Our ethics statement now only appears in the Methods section of the manuscript. The ethics statement that appeared on the title page has now been removed.

-> Response: We have reviewed our reference list and have made no changes. 

Reponse to Reviewer #1 (point 1-6): 

1. Is the manuscript technically sound, and do the data support the conclusions?

Reviewer #1: Yes

-> Response: Thank you.

2. Has the statistical analysis been performed appropriately and rigorously?

Reviewer #1: Yes

-> Response: Thank you.

3. Have the authors made all data underlying the findings in their manuscript fully available?

Reviewer #1: Yes

-> Response: we agree. Specifically, we have specified our data sharing policy and reason herefor above and in the manuscript.

4. Is the manuscript presented in an intelligible fashion and written in standard English?

Reviewer #1: Yes

-> Response: Thank you.

5. Review Comments to the Author

Reviewer #1: I've read with interest the paper by De Knegt et al. about the impact of the COVID-19 lockdown on birthweight among singleton term birth in Denmark. The study follows the scientific rigor and the manuscript is well written.

I have only one comment, Would the results be the same if the authors use local birthweight charts?

-> Response: We thank the reviewer for their question. In this study, we defined SGA as birth weight less than the 10th percentile and LGA greater than the 90th percentile at each completed week, for each sex, according to the international standards of newborn size from the INTERGROWTH-21st Project. Data from the INTERGROWTH 21st Project stem from eight different urban populations, and allows for comparisons across multiethnic populations. 

To ensure accuracy and appropriateness within our population, we also defined SGA and LGA as births with birth weight, corrected for sex and gestational week, below the 5th percentile or above the 95th percentile of the study population from 2015 - 2019, respectively. This thereby serves as an internal verification using local, Danish data.

We decided against using e.g. the Scandinavian intrauterine growth curves developed by Marsal et. Al. in 1996. These growth curves are intrauterine curves based on ultrasound-estimated foetal weights in women in Denmark and Sweden. We do not feel that these growth curves constitute appropriate local growth curves for birth weight. Back in 1996, ca. 24% of Danish and Swedish women were smokers. Today between 5-8% of pregnant women smoke. Furthermore, pregnant women living in Denmark today are more ethnically heterogenous than 26-years ago. Finally, the curves created by Marsal et al. were intended to understand intrauterine growth, not birth weight. No other appropriate local growth curves exist, and as such, the curves from the INTERGROWTH 21st Project as well as our own internal reference for LGA and SGA seem most appropriate. 

Regarding the wish of a reference to local weight charts, we believe we have met this as we refer to 95-percentile values from our own population - after having established that there was no significant change from 2015-2019 (table 3). Our findings regarding an increase in the proportion of LGA infants was apparent using both the INTERGROWTH 21 curves and our own internal cut-off percentiles from our own population. 

To clarify this, we have added a sentence to the manuscript to address the concern of not using a local/Danish growth curves: Line 281-284: ‘Furthermore, our approach of validating our findings using the international reference from the INTERGROWTH 21st Project by also referring to 95-percentile values for SGA and LGA in our own population addresses any issues concerning the lack of use of more local/Danish growth curves.’

6. PLOS authors have the option to publish the peer review history of their article (what does this mean?). If published, this will include your full peer review and any attached files.

Do you want your identity to be public for this peer review? For information about this choice, including consent withdrawal, please see our Privacy Policy.

Reviewer #1: No

---

## [Decision Letter · Decision Letter 1]

21 Mar 2023

The impact of the COVID-19 lockdown on birthweight among singleton term birth in DenmarkThe impact of the COVID-19 lockdown on birthweight among singleton term birth in Denmark

PONE-D-22-31109R1

Dear Dr. Lausten-Thomsen,

We’re pleased to inform you that your manuscript has been judged scientifically suitable for publication and will be formally accepted for publication once it meets all outstanding technical requirements.

Kind regards,

Andrea Maugeri

Academic Editor

PLOS ONE

Additional Editor Comments (optional):

Reviewers' comments:

Reviewer's Responses to Questions

**Comments to the Author**

1. If the authors have adequately addressed your comments raised in a previous round of review and you feel that this manuscript is now acceptable for publication, you may indicate that here to bypass the “Comments to the Author” section, enter your conflict of interest statement in the “Confidential to Editor” section, and submit your "Accept" recommendation.

Reviewer #1: All comments have been addressed

Reviewer #2: All comments have been addressed

2. Is the manuscript technically sound, and do the data support the conclusions?

Reviewer #1: Yes

Reviewer #2: Yes

3. Has the statistical analysis been performed appropriately and rigorously? 

Reviewer #1: Yes

Reviewer #2: Yes

4. Have the authors made all data underlying the findings in their manuscript fully available?

Reviewer #1: Yes

Reviewer #2: Yes

5. Is the manuscript presented in an intelligible fashion and written in standard English?

Reviewer #1: Yes

Reviewer #2: Yes

6. Review Comments to the Author

Reviewer #1: The authors have addressed appropriately my comments. The sentence added about local curves in the manuscript is sufficient.

Reviewer #2: The authors aimed to study the indirect effects of the COVID-19 pandemic on maternal and infant health examining birth weight and the relative-size-for-gestational-age-birth weight distribution during the initial lockdown implemented in Denmark compared to the previous five years.

This is a well-conceived and well-written paper including a large number of pregnancies with clean and complete data.

The first review leads to only one question that the authors answered correctly. I have no further comment.

7. PLOS authors have the option to publish the peer review history of their article (what does this mean?). If published, this will include your full peer review and any attached files.

Reviewer #1: No

Reviewer #2: No

---

## [Editor Report · Acceptance letter]

10 Apr 2023

PONE-D-22-31109R1 

The impact of the COVID-19 lockdown on birthweight among singleton term birth in Denmark 

Dear Dr. Lausten-Thomsen:

I'm pleased to inform you that your manuscript has been deemed suitable for publication in PLOS ONE. Congratulations! Your manuscript is now with our production department. 

Kind regards, 

on behalf of

Dr. Andrea Maugeri 

Academic Editor

PLOS ONE